# Exploring the Impact of Mindful Leadership on Employee Green Creativity in Manufacturing Firms: A Social Information Processing Perspective

**DOI:** 10.3390/bs14080712

**Published:** 2024-08-14

**Authors:** Baiqing Sun, Yuze Xi

**Affiliations:** School of Management, Harbin Institute of Technology, Harbin 150001, China; baiqingsun@hit.edu.cn

**Keywords:** mindful leadership, employee green creativity, moral reflectiveness, environmental passion, manufacturing

## Abstract

Increasingly, manufacturing enterprises are compelled to pursue innovative solutions to environmental issues. Addressing such issues requires mindful leadership to support employees and organizations in maintaining awareness during complex situations, which then promotes environmental sustainability. Drawing on social information processing theory, this study theorizes and tests the relationships between mindful leadership, employee moral reflectiveness, environmental passion, and employee green creativity. To test the model, we utilize a two-wave multisource dataset comprising 215 workers from manufacturing firms in China. The findings demonstrate that mindful leadership has a positive relationship with employee green creativity. Additionally, our research reveals that both moral reflectiveness and environmental passion serve as mediators in linking mindful leadership to employee green creativity. Crucially, our empirical analysis suggests a serial mediation model, examining the sequential role of moral reflectiveness and environmental passion in the relationship between mindful leadership and employee green creativity. The findings provide a new viewpoint on employees’ eco-friendly behaviors and have practical implications for improving environmental sustainability within organizations.

## 1. Introduction

Although vital for the economy, manufacturing companies face growing scrutiny because of their environmental impact [1,2], which increasingly pressures them to enhance their environmental practices in response to current environmental challenges and rising stakeholder demands [3]. The growing focus on environmental issues underscores the importance of creativity [4], which involves generating and implementing innovative ideas and strategies, and challenging conventional methods and entrenched organizational behaviors [5]. Creative behaviors vary depending on the desired results, and traditional creative outcomes do not apply directly to solving environmental issues [6]. Therefore, scholars have put forth the notion of green creativity, which involves the development of novel ideas regarding environmentally friendly products, services, processing methods, or practices that are seen as innovative, original, and practical [7]. This framework enables individuals to direct their attention toward a particular form of creativity dedicated to tackling environmental challenges, rather than being confined to a broader professional setting [8]. 

Employee green creativity plays a crucial role in manufacturing enterprises, significantly enhancing the organization’s long-term development and conferring numerous competitive advantages [9,10]. The innovative and environmentally friendly ideas generated by employees are instrumental in developing new green products and processes, which subsequently reduce negative impacts on the environment. Furthermore, the marketplace has experienced a burgeoning demand for environmentally sustainable products [11]. Employees endowed with green creativity can assist enterprises in developing products that offer environmental advantages, thereby securing a competitive edge in the market [12]. Moreover, the implementation of green creativity can promote green practices, create memorable customer experiences, and enhance both organizational value and customer value [13]. Building on these foundations, the study of employee green creativity has sparked significant enthusiasm and garnered substantial attention from scholars. Research has extensively examined the antecedents of employee green creativity at individual and organizational levels, including green human resource management [9,14], green transformational leadership [15], environment-specific servant leadership [16,17], green inclusive leadership [18], environment-specific empowering leadership [19], authentic leadership [20], corporate environmental ethics [21], environmental values [22], green self-efficacy [23], and green shared value [24].

In organizational contexts, leadership styles are increasingly recognized by academics as potential facilitators of environmental initiatives [25]; however, research remains limited, given that most studies concentrate on either conventional leadership or specifically green leadership. Research has provided crucial insights into the impact of leadership on employee creativity [15,16,17,18,19,20]. In the rapidly evolving business environment of today, leaders are of paramount importance. To address the uncertainty and complexity inherent in global environmental challenges, leaders must possess heightened sensitivity and vigilance, the ability to skillfully process dynamic information, and the competence to mitigate tensions effectively [26,27].

Mindful leadership—that is, utilizing mindful strategies to embrace change and navigate uncertainty—has garnered significant attention in organizational research [28,29,30]. This leadership approach underscores not only the personal development of leaders but also the application of mindfulness to cultivate a work environment conducive to employee growth and innovation [28,29]. Mindful leaders maintain a heightened awareness of the current context and environment [31] by facilitating improved focus and an accurate perception of evolving circumstances among their followers, relative to different scenarios [32,33]. Mindful leaders are crucial in shaping employees’ environmentally friendly behavior. In alignment with this, various positive outcomes of mindful leadership have been explored, including the promotion of employee green behavior [34], employee innovative behavior [33], and turnover intention [31]. However, empirical evidence on the direct impact of mindful leadership on employees’ environmentally friendly creative outcomes is limited, highlighting the need for further research that explores this relationship.

Manufacturing firms compete on the global market, and depending on the sector of manufacturing, the greening of the production process is highly dependent on technological innovations and international agreements. This paper focuses on the role of the leadership style of Chinese manufacturing firms on employee attitudes (or behaviors) in the context of green creativity. In addition to technological developments, this aspect has the potential for additional value in the greening of production within companies. We aim to explore why and how mindful leadership influences employee green creativity. This connection is important because mindful leadership prioritizes environmental sustainability, present-moment awareness, sharp environmental perception, and the achievement of tangible results. This approach can encourage employees to participate in environmental improvement activities, potentially leading to broader positive environmental impacts [32].

Research on organizational leadership and employee green creativity has also underscored the need for an in-depth investigation into the psychological processes underlying employee green creativity [14]. According to social information processing (SIP) theory [35], leaders are acknowledged as pivotal sources of social information in the workplace. By providing normative and expected information, leaders influence how employees interpret and comprehend the social cues they receive. Normative information encompasses guidance on work practices and standards, while expected information includes the leaders’ expectations and requirements of their team members [35]. This process enables employees to develop an understanding of their work environment, thereby significantly shaping their attitudes and perceptions [35]. Thus, this study proposes that mindful leadership affects employees’ cognition and emotions through a sequential process—moral reflectiveness and environmental passion—ultimately enhancing employee green creativity.

We propose that moral reflectiveness may mediate the relationship between mindful leadership and employee green creativity. Moral reflectiveness involves individuals actively engaging in introspective thought guided by moral principles, applying these insights to their daily interactions and experiences [36]. This reflective process prompts individuals to contemplate ethical implications and moral values as they navigate diverse situations and make decisions [37]. We argue that within the context of mindful leadership, moral reflectiveness plays a crucial role by enabling employees to interpret and respond to the signals and guidance provided by such leadership. This interpretation, we hypothesize, may significantly influence employees’ propensity for green creativity. By exploring these moral dimensions that have the potential to motivate employees toward green creativity, our research introduces a fresh theoretical perspective.

Moreover, we also argue that mindful leadership could affect employee green creativity by recognizing green information in the organization and achieving high green passion. Mindful leaders, as demonstrated by their keen awareness of environmental changes and proactive interest in sustainability efforts [31], actively engage in meaningful interactions with employees. This approach fosters open communication and cultivates strong, positive connections within the organizational context [38]. Additionally, a sense of optimism plays a crucial role in igniting enthusiasm and commitment toward environmental passion [39]. When employees actively identify and feel connected with their work environment, it inspires a passion for environmental sustainability, particularly among those who are well-informed about environmental degradation [40]. Strong passion enhances individuals’ energy, motivation, and inspiration, broadening their focus and cognitive flexibility to seize opportunities and innovate effectively [41]. We explore how mindful leadership can influences employees’ green passion and contribute to enhancing the green creativity of manufacturing companies. We also seek to examine how mindful leadership influences employee green creativity in a sequential manner, in which moral reflectiveness and environmental passion serve as serial mediators, specifically within the manufacturing industry in China.

This research represents a significant advancement in the scholarly landscape, offering notable contributions that enrich our understanding of key dynamics. First, by drawing from the framework of SIP theory, this research has investigated the role of mindful leadership as a pivotal precursor to employee green creativity, offering a fresh perspective on how this leadership style can bolster employee environmentally friendly creative outcomes. The discovery will support Rupprecht et al.’s (2019) assertion that mindful leadership can profoundly influence employee conduct, thereby reinforcing its role in predicting and evaluating outcomes [32]. Second, we explore the SIP theory of moral reflectiveness and environmental passion through which employee green creativity is influenced by mindful leadership, broadening comprehension of the reasons behind how mindful leadership stimulates employee green creativity. Finally, our study breaks new ground by exploring the sequential mediation pathway from mindful leadership to employee green creativity that is facilitated by moral reflectiveness and environmental passion. This intricate analysis elucidates the interconnected pathways through which mindful leadership exerts influence on employee green creativity, providing a comprehensive understanding of this multifaceted relationship.

## 2. Literature Review

### 2.1. Mindful Leadership and Employee Green Creativity

Mindfulness, originally derived from Buddhism, is an important Buddhist practice [42]. When it is introduced and applied in the field of organizational management, it is described as a keen awareness and acceptance of current events and experiences [26]. The effectiveness of mindfulness in leader development is supported by empirical research showing its positive effects on various aspects of leadership, personal development, and organizational outcomes [43,44]. As a result, it has gained recognition in the recent literature as a valuable tool for enhancing leadership effectiveness in today’s complex and demanding environments [43,44]. Mindful leadership is characterized by a leader’s consistent practice of maintaining heightened attention, prioritizing the present environment, carefully evaluating and making decisions on various issues, and guiding employees toward achieving mindfulness [32]. This approach emphasizes a leader’s active engagement with their surroundings, thoughtful consideration of challenges, and support for cultivating mindfulness among team members, thereby promoting a more focused, aware, and resilient organizational culture [45]. Mindful leadership is characterized by leaders embodying mindfulness traits that influence employee attitudes and behaviors through heightened awareness [29,31,44]. Leaders skilled in mindfulness demonstrate keen perceptiveness, which enables rapid recognition of employees’ needs and proactive efforts to meet them, thereby enhancing discernment [46]. Moreover, mindful leadership involves reflective engagement through emphasizing current conditions and contextual factors [31], which is crucial amid organizational dynamics and evolving circumstances. The aforementioned research indicates that mindful leadership has the potential to influence employees’ cognition, attitudes, and behaviors to a significant extent.

This study contends that mindful leadership will greatly boost employee green creativity for several key reasons. First, mindful leadership involves consciously focusing on the present moment and responding intentionally to situations [47]. This approach can greatly benefit employees by encouraging them to cultivate mindfulness themselves. According to SIP theory [35], leaders are perceived as important sources of information because of their significant social status and influence within the organization. Employees will perceive leaders’ ideas, demeanor, and conduct according to this information, which forms the foundation of the employees’ understanding and actions [48]. Mindful leaders’ openness to change signals to employees the importance of adapting to environmental shifts, which fosters a shared understanding of environmental preservation [33,46]. As a result, employees feel inspired to participate in actions that support corporate environmental initiatives in the workplace [49]. Compared with environmental conservation behaviors, employee green creativity emphasizes the generation of novel ideas, solutions, and approaches that are environmentally friendly [50,51,52]. It focuses on fostering innovative thinking and creativity to develop strategies, products, and processes that contribute to environmental sustainability and address environmental challenges effectively [50]. Research has found that mindfulness can improve individual attention to new stimuli, broaden perspectives, and foster a deeper understanding of complexity within dynamic environments, all of which improve employee creative outcomes [53,54].

What is more, mindful leaders who prioritize the present moment and demonstrate open acceptance significantly contribute to cultivating an inclusive and secure workplace atmosphere for employees [55]. By attentively focusing on the current environment and embracing diverse perspectives, they foster genuine understanding and respect, making employees feel valued and safe. This approach also supports employees’ green-related interests and reduces their sense of environmental uncertainty [42], thereby enhancing their confidence in achieving organizational sustainability goals [56]. Furthermore, it encourages employees to challenge traditional values, generate new green ideas, and take initiative in contributing to sustainable practice [41,57,58]. Therefore, from the analysis above, we propose the following hypothesis:

**Hypothesis** **1.**
*Mindful leadership positively relates to employee green creativity.*


### 2.2. Mediating Effects of Moral Reflectiveness

SIP theory asserts that individuals’ behaviors are influenced significantly by the cues they perceive [35]. This theory characterizes perception as a “reflective process”, emphasizing how individuals interpret their surroundings [35]. Within this framework, the way environmental signals are interpreted under mindful leadership is considered a reflective process within the realm of leadership dynamics. In contrast, moral reflectiveness pertains to a cognitive process involving introspection on moral issues and consideration of the ethical implications of one’s actions and experiences [36]. It requires individuals to exercise self-regulation guided by their conscience [59]. This conceptualization suggests that moral reflection includes a retrospective component in which individuals assess their past actions against personal moral principles. Empirical research supports moral reflection as a form of self-control [60] that enables individuals to adjust their behavior to uphold inherent moral standards. This paper considers moral reflectiveness as the intermediary that elucidates how mindful leadership influences employee green creativity.

Mindful leadership plays a pivotal role in fostering mindfulness and resilience among employees and organizations, particularly amid the complex challenges posed by global environmental uncertainties [28,32]. It emphasizes the cultivation of shared awareness and commitment to environmental conservation [33,46]. According to SIP theory [35], the information present within an organization has a strong impact on employees’ psychology and behavior. Through reflecting on leaders’ actions, employees enhance their understanding of ethical corporate conduct and contemplate their day-to-day moral choices within the organizational framework [49]. This reflective process enhances their moral responsibility and prompts thoughtful contemplation guided by ethical principles. Employees demonstrating a strong sense of responsibility are inclined to engage actively in moral introspection, thereby fostering moral reflectiveness [61]. Thus, mindful leadership encourages employees to engage in conscientious reflection guided by moral considerations.

We also suggest that employees who engage in moral reflectiveness are more likely to exhibit green creativity. People’s experiences are shaped typically by their foundational moral convictions because these beliefs serve as a framework for interpreting and reacting to various situations [62]. Moral beliefs influence how individuals perceive right and wrong, make decisions, and prioritize actions, thereby shaping their experiences and interactions with the world around them. Personal ethics fuel individuals’ commitments and enthusiasm toward environmental issues, highlighting moral contemplation as a crucial driver of pro-environmental actions [63]. These beliefs act as a lens through which people view their surroundings, guiding their behavior and responses in consistent ways. Employees who engage in moral reflection are more inclined to prioritize ethics by internalizing their moral systems and beliefs, which translates into positive workplace behaviors [64]. Employees with a high moral reflectiveness recognize the importance of environmental protection and sustainability, making them more aware of the impact of their actions on the environment [65]. This heightened awareness drives employees to take on environmental responsibilities more willingly by participating in green production and environmental protection activities. Through moral reflectiveness, the employees of manufacturing enterprises can realize the importance of green production more deeply and thus seek out and implement more environmentally friendly solutions in their daily work and promote the development of green creativity. Therefore, we posit the following hypothesis:

**Hypothesis** **2.**
*Moral reflectiveness mediates the positive relationship between mindful leadership and employee green creativity.*


### 2.3. Mediating Effects of Environmental Passion

Passion in employees can be described as a state of wholehearted dedication to pursuing specific tasks and activities persistently [66]. An individual with high passion can evoke feelings of pleasure and pride, compelling others to participate actively in specific behaviors [67]. The concept of environmental passion was thus proposed to increase employees’ awareness of and participation in environmental protection and sustainable development so that they might contribute to a more effective response to global environmental issues and achieve the goal of sustainable development [39]. Environmental passion denotes a steadfast emotional commitment and enthusiasm toward protecting and preserving the natural environment and is rooted in a profound interest in and dedication to sustainability and eco-friendly practices [39,41,68,69]. Environmental passion is considered a key driver of employees’ green outcomes [70,71,72]. In other words, employees who are passionate about the environment tend to be more motivated to fulfill the sustainability initiatives mandated by their organization and supervisors [15]. Studies have emphasized that the environmental passion of employees can be affected by leadership and organizational factors, such as environmental corporate social responsibility [67], green human resource management practices [15], and environment-specific transformational leadership [15]. Moreover, the environmental passion of employees can enhance their green intentions and behaviors, such as pro-environmental behaviors [73], organizational citizenship behavior for the environment, and environmental performance [74]. Consequently, this research seeks to investigate how environmental passion mediates the relationship between mindful leadership and employees’ environmentally friendly outcomes.

SIP theory [35] was utilized to investigate how mindful leadership influences employee environmental passion. First, through supervising employees in the workplace, leaders wield the authority to delegate tasks and engage in frequent daily interactions. These interactions serve as crucial cues for employees to interpret social information. Mindful leaders signal openness to change, enabling employees to comprehend environmental shifts and foster a collective commitment to environmental protection [33,46]. This shared understanding enhances employees’ alignments with organizational environmental initiatives [49], thereby boosting their passion for environmental issues. Furthermore, mindful leadership involves attentiveness to the concerns and status of team members, fostering trust and a sense of backing among employees toward both the leader and the company [31,32]. When employees feel supported by their organization, they are more open to messages related to environmental sustainability [17,33]. Exposure to such information enables employees to recognize the importance of environmental stewardship and develop a connection to their natural surroundings, and it fosters broader perspectives while fueling their passion for environmental initiatives [31,73].

We also propose that the environmental passion of employees sparks their green creativity. Passion from individuals fosters and motivates their enthusiastic engagements with and dedication to specific activities [66]. Strong commitment to environmental values is crucial for employees to participate actively in sustainable workplace practices. This passion fosters a positive mindset, motivating employees to embrace and promote green behavior. Environmental passion, which is characterized by deep affection and positive emotions toward environmental goals, serves as a significant motivator for employees, encouraging them to engage actively in challenging tasks [71]. This passion also plays a crucial role in fostering employees’ commitment to adopting environmentally conscious practices within the workplace [75]. Positive cognitive and emotional responses resulting from passion can broaden employees’ scope of thinking and actions, setting the stage for exploring unconventional approaches in their work tasks [54]. Furthermore, research indicates that feelings of excitement and energy propel individuals to be exploratory and creative [76]. Building upon the preceding points, we formulate the following hypothesis:

**Hypothesis** **3.**
*Environmental passion mediates the positive relationship between mindful leadership and employee green creativity.*


### 2.4. The Serial Mediating Role of Moral Reflectiveness and Environmental Passion in the Relationship between Mindful Leadership and Employee Green Creativity

Moral reflectiveness, which is defined as the depth to which individuals contemplate moral issues in their daily lives and decisions [36], plays a crucial role in fostering environmental passion. When employees engage in reflection on their moral beliefs and values, they gain a deeper understanding of the significance of environmental sustainability. Issues of social importance, which often resonate with personal values and societal concerns, can spark a sense of commitment and purpose [39]. This alignment between personal values and environmental goals enhances their passion for eco-friendly practices [63].

Moral reflectiveness is the degree to which individuals reflect on moral issues in their daily experiences and decisions [36]. First, when employees reflect on their moral beliefs and values, they are more likely to recognize the importance of environmental sustainability. Individuals are more likely to develop passion for issues of social importance because these issues often resonate deeply with personal values and societal concerns, fostering a sense of commitment and purpose [39]. This alignment between personal values and environmental goals can enhance their passion for eco-friendly practices [64]. Second, contemplating ethical values frequently fosters a sense of accountability for the common good, including environmental care, which can inspire employees to participate enthusiastically in eco-friendly practices. This point is supported by Kim et al. (2017) [61], who found that the prioritization of moral issues often leads individuals to see environmental actions as fulfilling their moral intentions. From the above, we hypothesize the following:

**Hypothesis** **4.**
*Mindful leadership can affect employee green creativity through the serial mediating roles of moral reflectiveness and environmental passion.*


To address these research questions, the current study initially formulated four hypotheses and constructed a serial mediation model, as illustrated in Figure 1.

## 3. Methods

### 3.1. Participants and Procedure

This research utilized a questionnaire-based survey approach. Questionnaire distribution and retrieval spanned from June 2024 to July 2024. Two waves of data were gathered from four manufacturing firms in China and covered machinery manufacturing, the automobile industry, and the fertilizer sector. We collaborated with human resource managers to compile a list of 300 randomly selected employees from a pool of over 600 employees and their respective supervisors. All participants were full-time working adults who participated online. Web-based questionnaires provide rapid distribution and extensive geographic coverage, mitigating the geographic constraints typically associated with traditional survey methods [77].

Various tactics were employed to address the issue of common method bias, as outlined by Podsakoff et al. (2012) [78]. First, we utilized established and validated scales from the existing literature. Second, prior to commencing the formal inquiry, we provided thorough guidance on questionnaire completion, stressing the importance of confidentiality and highlighting the academic nature of the survey. Third, we adopted a multisource approach. Employees reported individual-level variables such as mindful leadership, moral reflectiveness, and environmental passion, while data relating to employee green creativity were provided by employees’ immediate leaders.

A time-lagged survey-based design was used to examine our research model. In wave 1, we assessed several control variables—namely, age, gender, education, and job tenure—alongside mindful leadership, moral reflectiveness, and environmental passion. These variables were self-evaluated by the participating employees.

A total of 300 questionnaires were disseminated among participants, and upon completion of the survey period, 290 completed questionnaires were collected. Studies in the field have demonstrated that the introduction of a temporal gap between the measurement of the dependent variable and the criterion variables can serve as an effective strategy to alleviate potential bias stemming from common method variance [78]. A one-month time lag interval was selected because this study delved into an underlying psychological mechanism and its impact on employee green creativity. This timeframe aligns with research methodologies employed in similar studies [79]. Thus, one month later, employees’ immediate leaders were asked to rate employee green creativity. When individuals adhere to mindful leadership, it may lead to cognitive and emotional changes in their behaviors within a matter of weeks. Employee green creativity could consequently manifest during the second wave of data collection, coinciding with the period when employees are experiencing moral reflectiveness. Therefore, we employed two distinct surveys, spaced one month apart. A total of 290 questionnaires were distributed, of which 255 were retrieved. Following two waves of data collection and the subsequent removal of missing and unmatched data, a total of 215 complete datasets were retained. The response rate was 71.67%. Demographic profiles of participants are shown in Table 1.

### 3.2. Measurement

To guarantee the accuracy of the questionnaires, a translation–back translation procedure was employed [80]. This method involved translating the questionnaires into the target language, then translating them back into the original language to ensure consistency and fidelity of meaning [80]. The English questionnaire was translated into Chinese by a team of five bilingual Ph.D. candidates. Following this translation, a separate group of five lecturers undertook the task of translating the Chinese version back into English. The resulting back translation was assessed to ensure it accurately reflected the intended meaning of the original questionnaire. A 5-point Likert Scale was utilized to evaluate all measures in the study (1 = strongly disagree to 5 = strongly agree). After translation was completed, we established the scales, including mindful leadership, moral reflectiveness, environmental passion, and employee green creativity, and the control variables:

Mindful leadership: To assess mindful leadership, we employed a five-item scale that was initially developed by Schuh et al. (2019) [81]. A sample item was “My supervisor does jobs or tasks automatically, without being aware of what s/he is doing”. Cronbach’s alpha for this scale was 0.79.

Moral reflectiveness: We measured participants’ moral reflectiveness using Reynolds’s (2008) [47] 5-item scale. A sample item was “I value my own moral standards”. Cronbach’s alpha for this scale was 0.96.

Environmental passion: Ten items from the scale developed by Robertson and Barling (2013) [36] were used to measure environmental passion. A sample item read: “I am passionate about the environment”. Cronbach’s alpha was 0.92.

Employee green creativity: Six items from the scale developed by Chen and Chang (2013) [7] were used to measure employee green creativity. A sample item was “The members of the organization would rethink new green ideas”. Cronbach’s alpha was 0.85.

Control variables: To ensure consistency and accuracy in the analysis, factors such as age (21–30 years old = 1, 31–40 years old = 2, 41–50 years old = 3, older than 51 years old = 4), gender (male = 0, female = 1), education (under junior college = 1, junior college degrees = 2, bachelor’s degrees =3, master’s degrees or higher = 4), and the job tenure of employees (5 years or less = 1, 5–10 years = 2, 10–15 years = 3, 15 years or more = 4) were systematically controlled throughout the study. Previous research has indicated that these demographic variables can influence workplace behavior and may significantly affect individual green outcomes [82]. The demographic characteristics of the 291 valid participants are displayed in Table 1.

## 4. Results

### 4.1. Analytical Strategy and Preliminary Analysis

Statistical analysis was conducted using Mplus 7.0 and SPSS 26.0. In Harman’s single-factor analysis, a comprehensive examination revealed the presence of four distinct factors, each exhibiting eigenvalues surpassing the critical threshold of 1. Prior to any rotation, the initial factor’s variance interpretation rate stood at 34.26%, a figure within the acceptable range, not exceeding 50%. This indicates a proportionate level of variance captured by the initial factor structure, suggesting its adequacy for subsequent analytical procedures.

The study utilized Confirmatory Factor Analysis (CFA) via Mplus 7.0 software to evaluate how distinct the four key variables were empirically. The findings, which are shown in Table 2, indicate that the proposed four-factor model encompassing mindful leadership, moral reflectiveness, environmental passion, and employee green creativity, exhibit a satisfactory fit with the data (χ^2^ = 480.92; *df* = 293; *CFI* = 0.95; *TLI* = 0.94; *RMSEA* = 0.06; *SRMR* = 0.05).

Table 2 displays the fit indices for the three-factor models utilized in this research. Our findings reveal that the first three-factor model, which combined mindful leadership and moral reflectiveness into a single overarching factor, exhibited a poorer fit compared to the four-factor model (χ^2^ = 799.38; *df* = 296; *CFI* = 0.86; *TLI* = 0.84; *RMSEA* = 0.09; *SRMR* = 0.12). The findings for the second three-factor model, which integrated moral reflectiveness and environmental passion into a single overarching factor, similarly demonstrate an inferior fit compared to the four-factor mode (χ^2^ = 710.80; *df* = 296; *CFI* = 0.88; *TLI* = 0.87; *RMSEA* = 0.08; *SRMR* = 0.08). Similarly, the results for the third three-factor model, which amalgamated environmental passion and employee green creativity into a singular overarching factor, also exhibited a poorer fit compared to the four-factor model (χ^2^ = 635.82; *df* = 296; *CFI* = 0.90; *TLI* = 0.89; *RMSEA* = 0.07; *SRMR* = 0.07). The findings indicate that our model variables demonstrate sufficient discriminant validity, suggesting that they are effectively distinct from one another and measure unique constructs as intended.

### 4.2. Descriptive Statistics

Table 3 presents the mean, standard deviation, and correlation coefficient for the variables under investigation in this study, offering a comprehensive overview of their descriptive statistics and interrelationships.

Consistent with the arguments outlined earlier, there exists a positive association between mindful leadership and employee green creativity (r = 0.41, *p* < 0.01). Mindful leadership exerts a positive influence on employee moral reflectiveness (r = 0.16, *p* < 0.01) and environmental passion (r = 0.38, *p* < 0.01). Moral reflectiveness positively affects environmental passion (r = 0.34, *p* < 0.01) and employee green creativity (r = 0.41, *p* < 0.01). Environmental passion positively affects employee green creativity (r = 0.43, *p* < 0.01). The observed correlation aligns with the theoretical expectation, providing initial support for the hypothesis.

### 4.3. Hypothesis Testing

Hierarchical regression analysis, a statistical approach used to explore how variables interrelate while considering the influence of additional variables, was employed to evaluate Hypotheses 1 through 4. The regression outcomes, illustrating the associations between the variables and the hypotheses under investigation, are detailed in Table 4 and Table 5.

The findings of the study unveiled a significant correlation, indicating that individuals who show mindful leadership tend to exhibit higher levels of employee green creativity, as evidenced by a beta coefficient of 0.30 with a *p*-value less than 0.001, thereby lending empirical support to Hypothesis 1. Furthermore, Hypothesis 2, which proposed that moral reflectiveness serves as a partial mediator in the association between mindful leadership and employee green creativity, was validated. Moral reflectiveness positively influenced employee green creativity (*β* = 0.30, *p* < 0.001). Employee green creativity was positively influenced by the presence of mindful leadership (*β* = 0.35, *p* < 0.001). Even after incorporating moral reflectiveness into the regression model, the relationship between mindful leadership and employee green creativity remained significant (*β* = 0.30, *p* < 0.001). However, the strength of this effect diminished, indicating a weaker association compared to the initial analysis. Mackinnon et al. (2007) [83] have established the credibility of the distribution-of-the-product method for evaluating indirect effects. Employing SPSS PROCESS and opting for the fourth model, the analysis unveiled noteworthy findings. It demonstrated that moral reflectiveness played a significant mediating role in the relationship between mindful leadership and employee green creativity (*indirect effect* = 0.04, *SE* = 0.02, 95% *CI* = 0.01 to 0.10). Consequently, the evidence garnered from this study affirms Hypothesis 2, suggesting that the presence of moral reflectiveness indeed mediates the association between mindful leadership and employee green creativity.

Hypothesis 3, which posited that environmental passion acts as a partial mediator between mindful leadership and employee green creativity, received confirmation. Furthermore, as shown in Table 5, there was a positive impact of environmental passion on employee green creativity (*β* = 0.29, *p* < 0.001), and mindful leadership positively affected employee green creativity (*β* = 0.35, *p* < 0.001). Even after incorporating environmental passion into the regression model, the relationship between mindful leadership and employee green creativity remained significant (*β* = 0.24, *p* < 0.001). However, the strength of this effect diminished, indicating a weaker association compared to the initial analysis. By utilizing the SPSS PROCESS tool and selecting the fourth model, the analysis revealed significant mediation effects of environmental passion on the relationship between mindful leadership and employee green creativity (*indirect effect* = 0.07, *SE* = 0.04, 95% *CI =* [0.01, 0.15]). Consequently, Hypothesis 3 is affirmed.

Hypothesis 4, which proposed that moral reflectiveness and environmental passion serially mediate the relationship between mindful leadership and employee green creativity, was validated. The serial mediation model, linking mindful leadership with employee green creativity, was tested using Model 6 of SPSS PROCESS, a method advocated by Hayes (2022) [84] for its comprehensive evaluation of mediating pathways in statistical analyses.

The results in Table 6 showed that moral reflectiveness was identified as a mediator in the association between mindful leadership and employee green creativity, with a conditional indirect effect of 0.04 (*SE* = 0.02, 95% *CI* [0.01, 0.10]). Similarly, environmental passion was also found to mediate this relationship, exhibiting a conditional indirect effect of 0.07 (*SE* = 0.04, 95% *CI* [0.01, 0.15]). Furthermore, the results provided evidence for a serial mediating effect, indicating that both moral reflectiveness and environmental passion sequentially mediate the relationship between mindful leadership and employee green creativity, with a conditional indirect effect of 0.01 (*SE* = 0.01, 95% *CI* [0.001, 0.03]). Consequently, these findings collectively lend support to Hypothesis 4, underscoring the intricate interplay between mindful leadership, moral reflectiveness, environmental passion, and employee green creativity.

## 5. Discussion

As environmental challenges become increasingly significant and unavoidable for manufacturing enterprises, promoting employee green creativity is critical. This effort helps these firms to build a competitive advantage in sustainability by enhancing employees’ ability, motivation, and opportunities to generate new green ideas. The present study aimed to analyze how mindful leadership is linked to employee green creativity. Specifically, we examined the mediating role of moral reflectiveness and environmental passion in the relationship between mindful leadership and employee green creativity. As anticipated, the findings support all of the research hypotheses.

### 5.1. Theoretical Implications

This study carries significant theoretical implications. First, this study sheds light on a previously unrecognized positive connection between mindful leadership and employee green creativity in the manufacturing industry. Research has highlighted the increasing demand for eco-innovation and the acknowledgment of its potential benefits, prompting scholars to intensify their focus on understanding the drivers of eco-innovation and strategies for corporate involvement in this area [85]. There is a scarcity of research examining internal dynamics, such as the contributions of leaders and employees to fostering eco-innovation [86]. Drawing upon SIP theory [35], our research demonstrates that mindful leadership initiates interpretation processes of social information and normative and expected information, thereby influencing individuals’ attitudes and behaviors and significantly boosting employee green creativity. This finding provides empirical support for the notion that by fostering employees’ creative outcomes, mindful leadership is a potent organizational strategy for addressing the mounting environmental challenges [28,32]. Other empirical studies also conclude that mindful leadership positively influences employee work behavior [31,33]. This study adds to the mindful leadership literature by addressing the gap caused by a heavy focus on theoretical discussions of mindful leadership [28,32]; meanwhile, recent studies have underscored the importance of evaluating green creativity within the manufacturing sector [87]. To address this gap, our research proposes a framework for understanding green creativity in the manufacturing industry and responds to calls by Jia et al. (2018) [41] and Khalil and Abdallah (2023) [46] to expound on the antecedents of green creativity.

Second, according to the results of this study, moral reflectiveness mediates the relationship between mindful leadership and employee green creativity. This discovery significantly enriches our understanding of how mindful leadership affects employees’ environmental outcomes according to SIP theory. SIP theory assumes that individuals encode and interpret cues from the social environment that then influence their attitudes and behaviors [35]. Leadership critically shapes how employees interpret green information, which affects their green psychology and behavior [17]. Mindful leadership helps employees decrease the sense of unease caused by environmental issues and can create a common cognition of environmental protection in the workplace [32]. Such cues obtained by employees through a reflective process of leader behaviors can help them grasp the significance of environmental issues, prompting reflection on their daily experiences and moral decisions. Personal moral norms significantly influence pro-environmental behaviors [88]. Moral reflectiveness plays a pivotal role closely tied to the outcomes influenced by the environment [37,64]. This study stands out as one of the few to explore mediating effects in a green context, examining the relationship between mindful leadership and employees’ environmental outcomes. The theme is expanded upon by suggesting that moral reflectiveness serves as the primary motivating factor driving employee creativity in green initiatives.

Third, this study confirmed that environmental passion mediates the connection between mindful leadership and employee green creativity. While environmental passion is recognized as a critical component of green outcomes [15,70,71], research has not examined how environmental passion mediates the relationship between green leadership styles and employee green creativity. This gap in the literature highlights a promising avenue for future research. According to SIP theory [35], leaders influence employee attitudes, cognition, and behaviors through social norms and expectations in workplaces. Mindful leaders, who are aware of the current situation [31], enable their followers to perceive and respond better to changing conditions [32,43]. When employees see their leaders actively engaging in and prioritizing environmental responsibility, they are more likely to develop a passion for similar initiatives. This finding will offer valuable insights for creating strategies and policies that foster employees’ passion for environmental initiatives.

Fourth, our study uncovered a noteworthy positive serial mediation effect, in which moral reflectiveness and environmental passion jointly mediate the relationship between mindful leadership and employee green creativity. Studies have confirmed the mediation mechanism of employee cognitive, emotional, and behavioral factors, leadership, and green creativity [10]. By considering the sequential mediating effect of moral reflectiveness and environmental passion, we underscore the joint role of the cognitive factor (moral reflectiveness) and emotional component (environmental passion) of mindful leadership in driving the green creativity of employees. Our findings contribute to the understanding of how leadership style influences employee green creativity. They also extend the application of SIP theory within environmental management.

### 5.2. Practical Implications

Our findings carry specific practical implications that deserve attention. First, the findings revealed that mindful leadership is a key catalyst for employee green creativity, suggesting that organizations should focus on enhancing leaders’ mindfulness. We recommend that organizations implement structured mindfulness training programs for leaders. Organizations should also foster a workplace culture that values and supports mindfulness, encouraging all employees, including leaders, to participate in mindfulness activities and promote an environment of openness and presence. Furthermore, organizations can integrate mindfulness into manager selection criteria, prioritizing mindful leaders for environmental teams. These leaders should uphold mindfulness, emphasize environmental dynamics, assist in identifying conservation information, and encourage greener behaviors among employees.

Second, this study found that mindful leadership enhances the moral reflectiveness of employees, which in turn leads to increased employee green creativity. When practicing mindful leadership, leaders should pay close attention to the moral reflectiveness of their employees. Organizations can promote ethical behavior through leadership examples and support from top management. They can also create feedback mechanisms that enable employees to discuss ethical dilemmas and receive guidance. Research has found that employees’ pro-environmental outcomes are shaped by their moral characteristics [89]. Therefore, organizations can institute regular ethics training programs to enhance employees’ awareness and understanding of moral issues. Through utilizing training programs and environmental education workshops, organizations can bolster the environmental consciousness of their current employees, acknowledge environmental actions ethically, and foster an increase in employee green creativity.

Third, we found that mindful leadership can boost employee green creativity by strengthening employees’ environmental passion. These findings indicate that organizations can cultivate employees’ environmental passion by engaging them in the creation and execution of environmental initiatives. Hosting seminars, meetings, and workshops can increase awareness of environmental issues [41]. Providing education and training on environmental challenges and organizational goals helps employees internalize this information and feel aligned with an eco-conscious organization. Recognizing the impact of their organization’s environmental initiatives, particularly on the welfare of future generations, can motivate employees to advocate passionately for the environment.

## 6. Limitations and Future Research

Several limitations are present in this research that warrant acknowledgment. First, the study’s time-lagged research design collected data at two separate points. Despite measuring variables independently at different times, significant correlations can demonstrate the effectiveness of longitudinal analysis. Future research should employ a broader-scale survey, encompassing a more diverse array of manufacturing firms to authenticate and enhance our preliminary observations. Second, although we have collected data via two waves and research on employee green creativity has utilized similar measurement methods [90], more rigorous designs should be conducted to minimize concerns for common method bias (e.g., multisource design). Future research can also use longitudinal designs to examine how mindful leadership affects employee cognitive and emotional factors and innovative outcome behaviors over time. Third, our research was carried out within a single institutional setting, potentially limiting the generalizability of our findings to other national contexts. Research has shown that management practices and their impacts can exhibit considerable variation across different industries, sectors, and countries [91]. Last, beyond investigating how mindful leadership enhances employee green creativity through a chain mechanism, future research should also pinpoint other potential mediators and the boundary conditions that shape the relationship between mindful leadership and employee green creativity.

## 7. Conclusions

For this study, we utilized SIP theory as the conceptual framework to explore the relationship between mindful leadership and employee green creativity. A multilevel, multisource study was conducted to test our model in Chinese manufacturing companies. Our research presented a chain mediation model, which suggests that moral reflectiveness and environmental passion serve as mediators in the relationship between mindful leadership and employee green creativity. Understanding the intricate mechanisms that connect mindful leadership with employee green outcomes is essential for advancing ecological sustainability. Therefore, our research results provide a foundation for understanding the potential benefits of mindful leadership.

## Figures and Tables

**Figure 1 behavsci-14-00712-f001:**
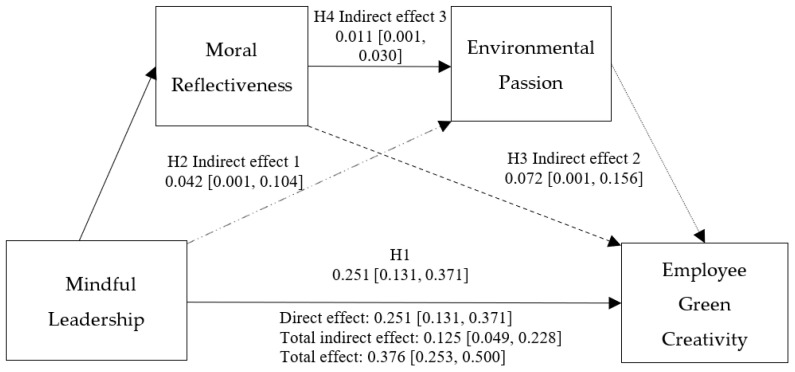
Theoretical model. Note: Direct effect 
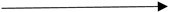
; Mediation Effect 1 
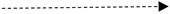
; Mediation Effect 2 
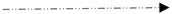
; Mediation Effect 3 
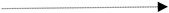
.

**Table 1 behavsci-14-00712-t001:** Demographic profile of participants.

Item	Category	Frequency	%
Gender	Male	123	57.21%
	Female	92	42.79%
Age	21–30	62	28.84%
	31–40	78	36.28%
	41–50	43	20.00%
	51 years or more	32	14.88%
Education	Under junior college	51	23.72%
	Junior college	83	38.60%
	Undergraduate	71	33.02%
	Master’s or above	10	4.65%
Job Tenure	5 years or fewer	57	26.51%
	5–10	48	22.33%
	10–15	64	29.77%
	15 years or more	46	21.40%

**Table 2 behavsci-14-00712-t002:** Results of confirmatory factor analysis.

Combination	χ^2^	*df*	*CFI*	*TLI*	*RMSEA*	*SRMR*
Four-factor model: ML, MR, EP, EGC	480.92	293	0.95	0.94	0.06	0.05
First three-factor model: ML + MR, EP, EGC	799.38	296	0.86	0.84	0.09	0.12
Second three-factor model: ML, MR + EP, EGC	710.80	296	0.88	0.87	0.08	0.08
Third three-factor model: ML, MR, EP + EGC	635.82	296	0.90	0.89	0.07	0.07

Note: ML = mindful leadership; MR = moral reflectiveness; EP = environmental passion; EGC = employee green creativity.

**Table 3 behavsci-14-00712-t003:** Descriptive statistics, correlations of variables.

	Mean	SD	1	2	3	4	5	6	7
1. Gender	0.57	0.50	-						
2. Age	2.21	1.02	−0.04						
3. Education	2.19	0.85	−0.73 **	−0.03					
4. Job tenure	2.46	1.10	−0.08	0.75 **	0.03				
5. Mindful leadership	4.09	0.60	−0.18 **	0.13 *	0.14 *	0.19 **			
6. Moral reflectiveness	3.63	1.10	−0.13	−0.17 *	0.15 *	−0.11	0.16 *		
7. Environmental passion	3.87	0.74	−0.14 *	0.06	0.11	0.13	0.38 **	0.34 **	
8. Employee green creativity	4.09	0.65	−0.39 **	−0.01	0.46 **	0.08	0.41 **	0.41 **	0.43 **

Note: n = 215. * *p* < 0.05, ** *p* < 0.01.

**Table 4 behavsci-14-00712-t004:** The mediating effect of moral reflectiveness.

	Moral Reflectiveness			Employee Green Creativity	
	Model 1	Model 2	Model 3	Model 4	Model 5
Control variables					
Gender	−0.07	−0.05	−0.11	−0.06	−0.05
Age	−0.19	−0.18	−0.10	−0.10	−0.05
Education	0.12	0.09	0.38 ***	0.36 ***	0.34 ***
Job tenure	0.01	−0.02	0.14	0.07	0.08
Independent variable					
Mindful leadership		0.16 *		0.35 ***	0.30 ***
Mediator					
Moral reflectiveness					0.30 ***
*R* ^2^	0.05	0.08	0.23	0.34	0.42
Adjusted *R*^2^	0.03	0.05	0.21	0.32	0.41
F	2.83 *	3.44 *	15.28 ***	21.50 ***	25.49 ***
Δ*R*^2^	0.05 *	0.03 *	0.23 ***	0.11 ***	0.08 ***

Note: n = 214. * *p* < 0.05, *** *p* < 0.001.

**Table 5 behavsci-14-00712-t005:** The mediating effect of environmental passion.

	Environmental Passion			Employee Green Creativity	
	Model 1	Model 2	Model 3	Model 4	Model 5
Control variables					
Gender	−0.12	−0.07	−0.11	−0.06	−0.04
Age	−0.09	−0.09	−0.10	−0.10	−0.08
Education	0.00	0.00	0.38 ***	0.36 ***	0.36 ***
Job tenure	0.19	0.13	0.14	0.07	0.04
Independent variable					
Mindful leadership		0.36 ***		0.35 ***	0.24 ***
Mediator					
Environmental passion					0.29 ***
*R* ^2^	0.04	0.16	0.23	0.34	0.41
Adjusted *R*^2^	0.02	0.14	0.21	0.32	0.39
F	2.05	7.86 ***	15.28 ***	21.50 ***	24.19 ***
Δ*R*^2^	0.04	0.12 ***	0.23 ***	0.11 ***	0.07 ***

Note: n = 215. *** *p* < 0.001.

**Table 6 behavsci-14-00712-t006:** Tests on serial mediation model of moral reflectiveness and environmental passion.

Model	Bootstrapping Results		95% Confidence Interval	
	Point Estimate	Boot SE	Boot LLCI	Boot ULCI
Total effects	0.376	0.062	0.253	0.500
Total direct effects	0.251	0.061	0.131	0.371
Total indirect effects	0.125	0.045	0.049	0.228
ML-MR-EGC	0.042	0.025	0.001	0.104
ML-MR-EGC	0.072	0.038	0.001	0.156
ML-MR-EP-EGC	0.011	0.008	0.001	0.030

Note: n = 215. Bootstrapping = 5000; ML = mindful leadership; MR = moral reflectiveness; EP = environmental passion; EGC = employee green creativity.

## Data Availability

The data that support the findings of this study are available from the corresponding author upon reasonable request.

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
