# Peer review of "Exploring the Impact of Mindful Leadership on Employee Green Creativity in Manufacturing Firms: A Social Information Processing Perspective"

_behavsci, 2024, doi:10.3390/bs14080712_

Round 1

Reviewer 1 Report

Comments and Suggestions for Authors

The approach of this paper is original. All the main elements needed are already present, but some details can still be easily improved. 

It would be helpful for the reader if you could frame somewhat your research question in a broader context to make the reader more easily to understand the scope of this particular paper (e.g. like this): manufacturing firms compete on the global market and depending on the sector of manufacturing, greening of the production process is highly dependent on technological innovations and international agreements. This paper focuses on the role of leadership style of Chinese manufacturing firms and the role of this on employee attitudes (or behaviour) in the sense of green creativity. In addition to technological developments, this aspect potentially has some additional value in the greening of production within companies. 

You could also describe generally, what kind of companies the respondents are working for? What type of manufacturing sectors, company size etc. Now this is very much a black box for the reader. 

In figure 1, the reader does not yet know about your empirical approach, so my recommendation is that you present it here first without any numerical information on the interrelationship between the different variables. You can present the same figure including the numerical information after the empirical analysis. 

rows 335-40: your method in surveying is quite good!

row 363- you mention that the employees´ immediate superiors were asked to rate employee green creativity. This raises some questions: 1) is this breaking the confidentiality of the employees´ survey? 2) there could easily be a bias, e.g. in case the superior could have discussed about the survey with the employee between the 1st and 2nd waves? You could clarify these issues a bit.

In the questionnaire the wording on mindful leadership is a bit odd: "...does automatically...". I understand that is is not possible to change the wording arterwards, but you could discuss the exact wording e.g. in a footnote.

row 444-, the correlations reported in the text do not match with the table, but they are mixed on the part of mindful leadership. Please check this. 

Taking the relatively limited number of replies in the questionnaire into account, you could discuss a bit whether these results give some support for your theoretical hypotheses or also, do you consider that the results represent something more general in the Chinese manufacturing companies?

Some minor issues:

-row 183-5: the verb seems to be missing from this sentence?

-row 353 tenure, you mean job tenure? For clarity, it would be better to use job tenure   

Comments on the Quality of English Language

minor editing and checking is enough

Author Response

Responses to Reviewer

Comments 1: It would be helpful for the reader if you could frame somewhat your research question in a broader context to make the reader more easily to understand the scope of this particular paper (e.g. like this): manufacturing firms compete on the global market and depending on the sector of manufacturing, greening of the production process is highly dependent on technological innovations and international agreements. This paper focuses on the role of leadership style of Chinese manufacturing firms and the role of this on employee attitudes (or behaviour) in the sense of green creativity. In addition to technological developments, this aspect potentially has some additional value in the greening of production within companies. 

Response 1: Thank you for providing constructive comments to our paper. We are glad that we have been granted an opportunity to revise and resubmit our paper. We have made further revisions to the paper in response to your comments. Below we detail the revisions we have made in response to the significant issues you’ve identified.  We add a general description of the research question in the introduction. Following is the relevant excerpt from the updated manuscript (revised sentences are marked by red color, line 81-87):

“Manufacturing firms compete on the global market, and depending on the sector of manufacturing, greening of the production process is highly dependent on technological innovations and international agreements. This paper focuses on the role of the leadership style of Chinese manufacturing firms on employee attitudes (or behaviors) in the context of green creativity. In addition to technological developments, this aspect has the potential for additional value in the greening of production within companies. We aim to explore why and how mindful leadership influences employee green creativity.”

Comments 2: You could also describe generally, what kind of companies the respondents are working for? What type of manufacturing sectors, company size etc. Now this is very much a black box for the reader.  

Response 2: Thanks for your comment and recommendation. We agree with you that a more comprehensive explanation of sample. Thus, we rewrote this part to show a more complete overview of sample. Following is the relevant excerpt from the updated manuscript (revised sentences are marked by red color, line 338-344):

“This research utilized a questionnaire-based survey approach. Questionnaire distribution and retrieval spanned from June 2024 to July 2024. Two waves of data were gathered from four manufacturing firms in China and covered machinery manufacturing, the automobile industry, and the fertilizer sector. We collaborated with human resource man-agers to compile a list of 300 randomly selected employees from a pool of over 600 employees and their respective supervisors. All participants were full-time working adults who participated online.”

Comments 3:  rows 335-40: your method in surveying is quite good!

Response 3: Thanks for your comment.

Comments 4:   row 363- you mention that the employees´ immediate superiors were asked to rate employee green creativity. This raises some questions: 1) is this breaking the confidentiality of the employees´ survey? 2) there could easily be a bias, e.g. in case the superior could have discussed about the survey with the employee between the 1st and 2nd waves? You could clarify these issues a bit.

Response 4: Thanks for this comment. First, we ensured that the employees' survey responses remained confidential by anonymizing all data before sharing it with their superiors for evaluation. This approach aimed to safeguard the privacy of individual responses while still allowing for an assessment of green creativity within the organizational context. Then, to minimize bias, we implemented several measures. We instructed superiors not to discuss the survey directly with their subordinates to prevent any influence on responses between the first and second waves of data collection. We also ask superiors to emphasize the importance of objective evaluation based on observable behaviors rather than personal interactions or discussions. These steps were taken to maintain the integrity of the research findings while balancing the need for comprehensive assessment of green creativity within the organizational hierarchy.

Comments 5:  In the questionnaire the wording on mindful leadership is a bit odd: "...does automatically...". I understand that is is not possible to change the wording arterwards, but you could discuss the exact wording e.g. in a footnote.

Response 5:  We acknowledge your observation regarding the wording of 'mindful leadership' in the questionnaire, specifically the phrase '...does automatically...'. While we cannot modify the questionnaire retrospectively, we appreciate your suggestion to clarify this phrasing. We will consider addressing this in our next communication or publication related to the research findings. Thank you for bringing this to our attention.

Comments 6:  row 444-, the correlations reported in the text do not match with the table, but they are mixed on the part of mindful leadership. Please check this.

Response 6 : Thanks for pointing out this issue. We will thoroughly review the data and ensure consistency between the text and the table. We appreciate your diligence in pointing this. We rewrote this part. Following is the relevant excerpt from the updated manuscript (revised sentences are marked by red color, line 446-1449):

“Consistent with the arguments outlined earlier, there exists a positive association between mindful leadership and employee green creativity (r = .41, p < .01). Mindful leadership exert a positive influence on employee moral reflectiveness (r=.16, p <.01) and environmental passion (r = .38, p < .01).”

Comments 7:  Taking the relatively limited number of replies in the questionnaire into account, you could discuss a bit whether these results give some support for your theoretical hypotheses or also, do you consider that the results represent something more general in the Chinese manufacturing companies?

Response: Thanks for pointing out this issue. The responses do provide some degree of support for our theoretical hypotheses. This suggests that our initial theoretical framework holds promise and warrants further investigation with a larger sample. Then, following your recommendation we rewrite limitation. Following is the relevant excerpt from the updated manuscript:

“Future research should employ a broader-scale survey, encompassing a more diverse array of manufacturing firms to authenticate and enhance our preliminary observations.”

Comments 8:  -row 183-5: the verb seems to be missing from this sentence?

Response 8: Thank you for encouraging us to clarify and rewrite this part. We are sorry for causing the confusion.  First, we checked the sentence carefully and added the verbs. Secondly, we have revised the language in the article. Finally, we hired a professional language embellishment company to check the language of the article. The attached document is a language embellishment certificate. Following is the relevant excerpt from the updated manuscript (please also see line 178-184 in the updated manuscript):

Employees will perceive leaders’ ideas, demeanor, and conduct according to this information, which forms the foundation of the employees’ understanding and actions [49]. Mindful leaders’ openness to change signals to employees the importance of adapting to environmental shifts, which fosters a shared understanding of environmental preservation [34, 47]. As a result, employees feel inspired to participate in actions that support corporate environmental initiatives in the workplace [50].”

Comments 9: -row 353 tenure, you mean job tenure? For clarity, it would be better to use job tenure.   

Response: Thanks for pointing out this issue. We use job tenure instead of tenure in the updated manuscript.

In sum, we appreciate your encouraging comments and your efforts of providing us with a set of detailed and helpful suggestions in this review. It is clear you have spent a significant amount of time and effort in offering such high-quality feedback. We believe that your suggestions have helped push our thinking further in terms of clarity and depth of contribution. Thank you so much!

Reviewer 2 Report

Comments and Suggestions for Authors

Dear Authors, thank you for interesting article.

The findings contribute to the understanding of how leadership style influences employee green creativity, while also extending the application of SIP theory within environmental management.

This is an interesting example of an interdisciplinary article using psychological concepts to explain organizational decision-making processes. 90 literature sources were used, most of which are works from recent years.

The appropriate methodological approach was used. The structure of the article is appropriate.

Author Response

Comments 1: Dear Authors, thank you for interesting article.

The findings contribute to the understanding of how leadership style influences employee green creativity, while also extending the application of SIP theory within environmental management. This is an interesting example of an interdisciplinary article using psychological concepts to explain organizational decision-making processes. 90 literature sources were used, most of which are works from recent years. The appropriate methodological approach was used. The structure of the article is appropriate.

Response 1: we are grateful for the time and effort you have committed to reviewing our paper and providing us constructive feedback. We genuinely believe that our manuscript has improved as a result of the review process.

Reviewer 3 Report

Comments and Suggestions for Authors

The topic is interesting, but I have some minor suggestions:

·        The methodology section also needs further development. Sampling technique was not mentioned. How did the author distribute the survey and where? It is therefore advised for the authors to state which one was used any why?

·        The authors stated that the data was collected from four manufacturing firms in China, but it is not clear from which sector. Are they all from the same sector? So, more detailed information should be provided.

·        The generated model from the software should be presented in the findings’ section.

·        Discussion section is very short. The author has discussed the results and supported them in theoretical implications section. Instead, all results of hypotheses should be discussed and supported by past studies in discussion section.

·        Proofreading is required as there are some grammatical errors.

Comments on the Quality of English Language

The topic is interesting, but I have some minor suggestions:

·        The methodology section also needs further development. Sampling technique was not mentioned. How did the author distribute the survey and where? It is therefore advised for the authors to state which one was used any why?

·        The authors stated that the data was collected from four manufacturing firms in China, but it is not clear from which sector. Are they all from the same sector? So, more detailed information should be provided.

·        The generated model from the software should be presented in the findings’ section.

·        Discussion section is very short. The author has discussed the results and supported them in theoretical implications section. Instead, all results of hypotheses should be discussed and supported by past studies in discussion section.

·        Proofreading is required as there are some grammatical errors.

Author Response

Comments 1: The topic is interesting, but I have some minor suggestions: The methodology section also needs further development. Sampling technique was not mentioned. How did the author distribute the survey and where? It is therefore advised for the authors to state which one was used any why? 

Response 1: Thank you for providing constructive comments to our paper. We are glad that we have been granted an opportunity to revise and resubmit our paper. We have made further revisions to the paper in response to your comments. Below we detail the revisions we have made in response to the significant issues you’ve identified.   

We add sampling technique (random). We distribute the survey online. Following is the relevant excerpt from the updated manuscript (revised sentences are marked by red color, line 338-375):

“This research utilized a questionnaire-based survey approach. Questionnaire distribution and retrieval spanned from June 2024 to July 2024. Two waves of data were gathered from four manufacturing firms in China and covered machinery manufacturing, the automobile industry, and the fertilizer sector. We collaborated with human resource man-agers to compile a list of 300 randomly selected employees from a pool of over 600 employees and their respective supervisors. All participants were full-time working adults who participated online. Web-based questionnaires provide rapid distribution and extensive geographic coverage, mitigating the geographic constraints typically associated with traditional survey methods [78].

 Various tactics were employed to address the issue of common method bias, as out-lined by Podsakoff et al. (2012) [79]. First, we utilized established and validated scales from existing literature. Second, prior to commencing the formal inquiry, we provided thorough guidance on questionnaire completion, stressing the importance of confidentiality and highlighting the academic nature of the survey. Third, we adopted a multisource approach. Employees reported individual-level variables such as mindful leadership, moral reflectiveness, and environmental passion, while data relating to employee green creativity were provided by employees’ immediate leaders.

A time-lagged survey-based design was used to examine our research model. In wave 1, we assessed several control variables—namely, age, gender, education, and job tenure—alongside mindful leadership, moral reflectiveness, and environmental passion. These variables were self-evaluated by the participating employees.

A total of 300 questionnaires were disseminated among participants, and upon completion of the survey period, 290 completed questionnaires were collected. Studies in the field have demonstrated that the introduction of a temporal gap between the measurement of the dependent variable and the criterion variables can serve as an effective strategy to alleviate potential bias stemming from common method variance [79]. A one-month time lag interval was selected because this study delved into an underlying psychological mechanism and its impact on employee green creativity. This timeframe aligns with re-search methodologies employed in similar studies [80]. Thus, one month later, employees’ immediate leaders were asked to rate employee green creativity. When individuals adhere to mindful leadership, it may lead to cognitive and emotional changes in their behaviors within a matter of weeks. Employee green creativity could consequently manifest during the second wave of data collection, coinciding with the period when employees are experiencing moral reflectiveness. Therefore, we employed two distinct surveys, spaced one month apart. A total of 290 questionnaires were distributed, of which 255 were retrieved. Following two waves of data collection and subsequent removal of missing and un-matched data, a total of 215 complete datasets were retained. The response rate was 71.67%. Demographic profiles of participants are shown in Table 1.”

Comments 2: The authors stated that the data was collected from four manufacturing firms in China, but it is not clear from which sector. Are they all from the same sector? So, more detailed information should be provided.  

Response 2: Thanks for your comment and recommendation. We agree with you that a more comprehensive explanation of sample. Thus, we rewrote this part to show a more complete overview of sample. Following is the relevant excerpt from the updated manuscript (revised sentences are marked by red color, line 339-341):

“Two waves of data were gathered from four manufacturing firms in China, covering machinery manufacturing, the automobile industry, and the fertilizer sector.”

Comments 3:  The generated model from the software should be presented in the findings’ section.

Response 3: Thanks for your comment and recommendation. Software is added in the findings’ section. Following is the relevant excerpt from the updated manuscript:

“Statistical analysis was conducted using Mplus 7.0 and SPSS 26.0.”

“The study utilized Confirmatory Factor Analysis (CFA) via Mplus 7.0 software to evaluate how distinct the four key variables were empirically.”

“Employing SPSS PROCESS and opting for the fourth model, the analysis unveiled note-worthy findings. It demonstrated that moral reflectiveness played a significant mediating role in the relationship between mindful leadership and employee green creativity.”

“By utilizing the SPSS PROCESS tool and selecting the fourth model, the analysis revealed significant mediation effects of environmental passion on the relationship between mindful leadership and employee green creativity.”

“The serial mediation model, linking mindful leadership with employee green creativity, was tested using Model 6 of SPSS PROCESS, a method advocated by Hayes (2022) [83] for its comprehensive evaluation of mediating pathways in statistical analyses.”

Comments 4:   Discussion section is very short. The author has discussed the results and supported them in theoretical implications section. Instead, all results of hypotheses should be discussed and supported by past studies in discussion section.

Response 4: Thanks for this comment. With your guidance, we have made the following corrections: First, we discussed the results and supported them in theoretical implications section. Then, all results of hypotheses are discussed and supported by past studies in discussion section. Following is the relevant excerpt from the updated manuscript (revised sentences are marked by red color, line 522-583):

“5.1. Theoretical Implications

This study carries significant theoretical implications. First, this study sheds light on a previously unrecognized positive connection between mindful leadership and employee green creativity in the manufacturing industry. Research has highlighted the increasing demand for eco-innovation and the acknowledgment of its potential benefits, prompting scholars to intensify their focus on understanding the drivers of eco-innovation and strategies for corporate involvement in this area [86]. There is a scarcity of research examining internal dynamics, such as the contributions of leaders and employees to fostering eco-innovation [87]. Drawing upon SIP theory [36], our research demonstrates that mindful leadership initiates interpretation processes of social information and normative and expected information, thereby influencing individuals’ attitudes and behaviors and significantly boosting employee green creativity. This finding provides empirical support for the notion that by fostering employees’ creative outcomes, mindful leadership is a potent organizational strategy for addressing the mounting environmental challenges [29,33]. Other empirical studies also conclude that mindful leadership positively influences employee work behavior [32,34]. This study adds to the mindful leadership literature by ad-dressing the gap caused by a heavy focus on theoretical discussions of mindful leadership [29,33]; meanwhile, recent studies have underscored the importance of evaluating green creativity within the manufacturing sector [88]. To address this gap, our research proposes a framework for understanding green creativity in the manufacturing industry and responds to calls by Jia et al. (2018) [42] and Khalil and Abdallah (2023) [47] to extend the antecedents of green creativity.

Second, according to the results of this study, moral reflectiveness mediates the relationship between mindful leadership and employee green creativity. This discovery significantly enriches our understanding of how mindful leadership affects employees’ environmental outcomes according to SIP theory. SIP theory assumes that individuals encode and interpret cues from the social environment that then influence their attitudes and behaviors [36]. Leadership critically shapes how employees interpret green information, which affects their green psychology and behavior [18]. Mindful leadership helps employees decrease the sense of unease caused by environmental issues and can create a common cognition of environmental protection in the workplace [33]. Such cues obtained by employees through a reflective process of leader behaviors can help them grasp the significance of environmental issues, prompting reflection on their daily experiences and moral decisions. Personal moral norms significantly influence pro environmental behaviors [89]. Moral reflectiveness plays a pivotal role closely tied to the outcomes influenced by the environment [38,65]. This study stands out as one of the few to explore mediating effects in a green context, examining the relationship between mindful leadership and employees’ environmental outcomes. The assertion is expanded upon by suggesting that moral reflectiveness serves as the primary motivating factor driving employee creativity in green initiatives.

Third, this study confirmed that environmental passion mediates the connection be-tween mindful leadership and employee green creativity. While environmental passion is recognized as a critical component of green outcomes [16,71,72], research has not examined how environmental passion mediates the relationship between green leadership styles and employee green creativity. This gap in the literature highlights a promising avenue for future research. According to SIP theory [36], leaders influence employee attitudes, cognition, and behaviors through social norms and expectations in workplaces. Mindful leaders, who are aware of the current situation [32], enable their followers to perceive and respond better to changing conditions [33,44]. When employees see their leaders actively engaging in and prioritizing environmental responsibility, they are more likely to develop a passion for similar initiatives. This finding will offer valuable insights for creating strategies and policies that foster employees’ passion for environmental initiatives.

Fourth, our study uncovered a noteworthy positive serial mediation effect, in which moral reflectiveness and environmental passion jointly mediate the relationship between mindful leadership and employee green creativity. Studies have confirmed the mediation mechanism of employee cognitive, emotional, and behavioral factors, leadership, and green creativity [10]. By considering the sequential mediating effect of moral reflectiveness and environmental passion, we underscore the joint role of the cognitive factor (moral reflectiveness) and emotional component (environmental passion) of mindful leadership in driving the green creativity of employees. Our findings contribute to the understanding of how leadership style influences employee green creativity. They also extend the application of SIP theory within environmental management.”

Comments 5:  Proofreading is required as there are some grammatical errors.

Response 5:  Thanks for your comment.  First, we have revised the language in the article. Second, we hired a professional language embellishment company to check the language of the article. The attached document is a language embellishment certificate.

In sum, we appreciate your encouraging comments and your efforts of providing us with a set of detailed and helpful suggestions in this review. It is clear you have spent a significant amount of time and effort in offering such high-quality feedback. We believe that your suggestions have helped push our thinking further in terms of clarity and depth of contribution. Thank you so much!

Reviewer 4 Report

Comments and Suggestions for Authors

This is an interesting article that explores the link between mindful leadership and employee green creativity. The research gap is well stated, and the research in general is well justified. However, there are some points that need attention:

1.      Line 98: The authors propose moral reflectiveness and environmental passion as mediating variables between mindful leadership and employee green creativity, and these concepts are considered for the theoretical framework. I think the authors should justify this theoretical model in more detail. This is because it is likely that there are other relevant factors that influence this relationship. For example, the opinions of employees' colleagues as revealed behavioural studies in other contexts, which is consistent with the principles of the SIP theory (i.e. how individuals interpret their surrounding). I am not saying that the authors should add additional factors. But the omission of some drivers of behaviour has to be acknowledged in the limitations.

2.      Line 144: The authors state the following: “To address these research questions, the current study initially formulated four hypotheses and constructed a serial mediation model, as illustrated in Figure 1”. This is confusing. The authors have not formally proposed research questions.

3.      Line 307: The authors state the following: “From the preceding analysis, it is evident that the mindful leadership can influence employee green creativity through two primary mechanisms: moral reflectiveness and environmental passion”. I think this is a very strong assumption. If what the authors explain is evident, then why to develop this research after all? The authors should propose these ideas as possibilities that will be confirmed with empirical data.

4.      Methodology: Please explain the sampling technique adopted in the research. If the sample is not random, mention this in the limitations.

5.      Lines 343 and 376: There is a repetition in these lines. That is, the authors explain that a process of translation-back translation was applied to the questionnaire. I don’t see the need for this repetition. Better to keep this information in Section 3.2.

6.      Line 382: The type of scale that the authors have adopted is referred to as 5-points Likert Scale. Please amend.

7.      Please explain and justify the statistical approach adopted in the research and the software in the methodological section. It is unusual explain this in the results section.

8.      Tables 2: Explain the main information contained in Table 2, after this table. No everyone will understand what the meaning of this information is. Please do the same with Table 3.

9.      The effect of mindful leadership on moral reflectiveness is significant at 5% of significant level. However, the coefficient of determination (R square) is extremely low (0.08). This suggests, as I explain in Point 1 above, that there are other omitted relevant drivers of behaviour. The problem with this, is that the results from the regression analysis may be biased because of variable omission (the technical name of this is omitted variable bias), and this means that the results obtained in the regressions have to be considered with caution. This is important because one of the key findings of this study may not hold. Authors should acknowledge this in the limitations.

10.  Conclusions are very poor. This section looks like an abstract. Please expand explaining the main implications of the research. Also, limitations and proposals for future research are normally placed at the end of the conclusions.

Author Response

Comments 1: This is an interesting article that explores the link between mindful leadership and employee green creativity. The research gap is well stated, and the research in general is well justified. However, there are some points that need attention:

 Line 98: The authors propose moral reflectiveness and environmental passion as mediating variables between mindful leadership and employee green creativity, and these concepts are considered for the theoretical framework. I think the authors should justify this theoretical model in more detail. This is because it is likely that there are other relevant factors that influence this relationship. For example, the opinions of employees' colleagues as revealed behavioural studies in other contexts, which is consistent with the principles of the SIP theory (i.e. how individuals interpret their surrounding). I am not saying that the authors should add additional factors. But the omission of some drivers of behaviour has to be acknowledged in the limitations.

Response 1: Thank you for providing constructive comments to our paper. We are glad that we have been granted an opportunity to revise and resubmit our paper. We have made further revisions to the paper in response to your comments. Below we detail the revisions we have made in response to the significant issues you’ve identified.  

Regarding the suggestion that there may be other relevant factors influencing this relationship, such as colleagues' opinions as per Social Information Processing (SIP) theory, we agree that these factors merit consideration. We did consider various potential mediators in future research. We will also address the limitations regarding the omission of certain drivers of behavior, ensuring transparency about the scope and boundaries of our theoretical framework.

 We rewrite limitation part. Following is the relevant excerpt from the updated manuscript (revised sentences are marked by red color, p. 631-635):

“Last, beyond investigating how mindful leadership enhances employee green creativity through a chain mechanism, future research should also pinpoint other potential mediators and the boundary conditions that shape the relationship between mindful leadership and employee green creativity.”

Comments 2: Line 144: The authors state the following: “To address these research questions, the current study initially formulated four hypotheses and constructed a serial mediation model, as illustrated in Figure 1”. This is confusing. The authors have not formally proposed research questions.  

Response 2: Thanks for your comment and recommendation. We removed this and put Figure 1 after Literature Review.

Comments 3:   Line 307: The authors state the following: “From the preceding analysis, it is evident that the mindful leadership can influence employee green creativity through two primary mechanisms: moral reflectiveness and environmental passion”. I think this is a very strong assumption. If what the authors explain is evident, then why to develop this research after all? The authors should propose these ideas as possibilities that will be confirmed with empirical data.

Response 3: Thanks for your comment and recommendation. We are very sorry that our language is not accurate enough, we have deleted this sentence.

Comments 4:  Methodology: Please explain the sampling technique adopted in the research. If the sample is not random, mention this in the limitations.

Response 4: Thanks for this comment. We add sampling technique (random). We distribute the survey online. Following is the relevant excerpt from the updated manuscript (revised sentences are marked by red color, line 338-346):

“This research utilized a questionnaire-based survey approach. Questionnaire distribution and retrieval spanned from June 2024 to July 2024. Two waves of data were gathered from four manufacturing firms in China and covered machinery manufacturing, the automobile industry, and the fertilizer sector. We collaborated with human resource managers to compile a list of 300 randomly selected employees from a pool of over 600 employees and their respective supervisors. All participants were full-time working adults who participated online. Web-based questionnaires provide rapid distribution and extensive geographic coverage, mitigating the geographic constraints typically associated with traditional survey methods [78].

Comments 5:  Lines 343 and 376: There is a repetition in these lines. That is, the authors explain that a process of translation-back translation was applied to the questionnaire. I don’t see the need for this repetition. Better to keep this information in Section 3.2.

Response 5:  Thanks for pointing out this issue. Following your recommendation, we delete the repetition, and keep this information in Section 3.2.

Comments 6:   Line 382: The type of scale that the authors have adopted is referred to as 5-points Likert Scale. Please amend.

Response 6: Thanks for pointing out this issue. We rewrote this part. Following is the relevant excerpt from the updated manuscript:

“5-points Likert Scale was utilized to evaluate all measures in the study (1 = strongly disagree to 5 = strongly agree).

Comments 7:  Please explain and justify the statistical approach adopted in the research and the software in the methodological section. It is unusual explain this in the results section.

Response 7: Thanks for your comment and recommendation. Software is added in the findings’ section. Following is the relevant excerpt from the updated manuscript:

“Statistical analysis was conducted using Mplus 7.0 and SPSS 26.0.”

“The study utilized Confirmatory Factor Analysis (CFA) via Mplus 7.0 software to evaluate how distinct the four key variables were empirically.”

“Employing SPSS PROCESS and opting for the fourth model, the analysis unveiled note-worthy findings. It demonstrated that moral reflectiveness played a significant mediating role in the relationship between mindful leadership and employee green creativity.”

“By utilizing the SPSS PROCESS tool and selecting the fourth model, the analysis revealed significant mediation effects of environmental passion on the relationship between mindful leadership and employee green creativity.”

“The serial mediation model, linking mindful leadership with employee green creativity, was tested using Model 6 of SPSS PROCESS, a method advocated by Hayes (2022) [83] for its comprehensive evaluation of mediating pathways in statistical analyses.”

Comments 8: Tables 2: Explain the main information contained in Table 2, after this table. No everyone will understand what the meaning of this information is. Please do the same with Table 3.

Response 8: Thank you for encouraging us to clarify and rewrite this part. We are sorry for causing the confusion.  Following is the relevant excerpt from the updated manuscript  (please also see line 413-437 in the updated manuscript):

The study utilized Confirmatory Factor Analysis (CFA) via Mplus 7.0 software to evaluate how distinct the four key variables were empirically. The findings, which is shown in table 2, indicated that the proposed four-factor model encompassing mindful leadership, moral reflectiveness, environmental passion, employee green creativity, exhibited a satisfactory fit with the data (χ2 = 480.92; df = 293; CFI = .95; TLI = .94; RMSEA = .06; SRMR = .05).

Table 2 displays the fit indices for the three-factor models utilized in this research. Findings revealed that the first three-factor model, which combined mindful leadership and moral reflectiveness into a single overarching factor, exhibited poorer fit compared to the four-factor model (χ2 = 799.38; df = 296; CFI = .86; TLI = .84; RMSEA = .09; SRMR = .12). The findings for the second three-factor model, which integrated moral reflectiveness and environmental passion into a single overarching factor, similarly demonstrated inferior fit compared to the four-factor mode (χ2 = 710.80; df = 296; CFI = .88; TLI = .87; RMSEA = .08; SRMR = .08). Similarly, the results for the third three-factor model, which amalgamated environmental passion and employee green creativity into a singular overarching factor, also exhibited poorer fit compared to the four-factor model (χ2 = 635.82; df = 296; CFI = .90; TLI = .89; RMSEA = .07; SRMR = .07). The findings indicate that our model variables demonstrate sufficient discriminant validity, suggesting that they are effectively distinct from one another and measure unique constructs as intended.”

“Table 3 presents the mean, standard deviation, and correlation coefficient for the variables under investigation in this study, offering a comprehensive overview of their descriptive statistics and interrelationships.

Consistent with the arguments outlined earlier, there exists a positive association between mindful leadership and employee green creativity (r = .41, p < .01). Mindful leadership exert a positive influence on employee moral reflectiveness (r=.16, p <.01) and environmental passion (r = .38, p < .01). Moral reflectiveness positively affects environmental passion (r= .34, p < .01) and employee green creativity (r = .41, p < .01). Environmental passion positively affects employee green creativity (r = .43, p < .01). The observed correlation aligns with the theoretical expectation, providing initial support for the hypothesis.”

Comments 9: The effect of mindful leadership on moral reflectiveness is significant at 5% of significant level. However, the coefficient of determination (R square) is extremely low (0.08). This suggests, as I explain in Point 1 above, that there are other omitted relevant drivers of behaviour. The problem with this, is that the results from the regression analysis may be biased because of variable omission (the technical name of this is omitted variable bias), and this means that the results obtained in the regressions have to be considered with caution. This is important because one of the key findings of this study may not hold. Authors should acknowledge this in the limitations. 

Response 9: Thanks for pointing out this issue. We aim to explore why and how mindful leadership influences employee green creativity based on SIP theory. Under the guidance of other theories, other explanation mechanisms can not be ruled out. In light of your valuable comments, we have added the contents of limitation as follows (please also see line 631-635 in the updated manuscript):

“Last, beyond investigating how mindful leadership enhances employee green creativity through a chain mechanism, future research should also pinpoint other potential media-tors and the boundary conditions that shape the relationship between mindful leadership and employee green creativity.”

Comments 10:  Conclusions are very poor. This section looks like an abstract. Please expand explaining the main implications of the research. Also, limitations and proposals for future research are normally placed at the end of the conclusions.

Response 10: Thanks for pointing out this issue.  Limitations and proposals for future research is shown in Limitations and Future Research part. We rewrite the this two part  (please also see line 618-645 in the updated manuscript):

“6. Limitations and Future Research

Several limitations are present in this research that warrant acknowledgment. First, the study’s time-lagged research design collected data at two separate points. Despite measuring variables independently at different times, significant correlations can effectively demonstrate longitudinal analysis. Future research should employ a broader-scale survey, encompassing a more diverse array of manufacturing firms to authenticate and enhance our preliminary observations. Second, although we have collected data via two waves and research on employee green creativity has utilized similar measurement methods [91], more rigorous designs should be conducted to minimize concerns for common method bias (e.g., multisource design). Future research can also use longitudinal designs to examine how mindful leadership affects employee cognitive and emotional factors and innovative outcome behavior over time. Third, our research was carried out within a single institutional setting, potentially limiting the generalizability of our findings to other national contexts. Research has shown that management practices and their impacts can exhibit considerable variation across different industries, sectors, and countries [92]. Last, beyond investigating how mindful leadership enhances employee green creativity through a chain mechanism, future research should also pinpoint other potential mediators and the boundary conditions that shape the relationship between mindful leadership and employee green creativity.

  1. Conclusions

For this study, we utilized SIP theory as the conceptual framework to explore the relationship between mindful leadership and employee green creativity. A multilevel, multi-source study was conducted to test our model in Chinese manufacturing companies. Our research presented a chain mediation model, which suggests that moral reflectiveness and environmental passion serve as mediators in the relationship between mindful leadership and employee green creativity. Understanding the intricate mechanisms that connect mindful leadership with employee green outcomes is essential for advancing ecological sustainability. Therefore, our research results provide a foundation for understanding the potential benefits of mindful leadership.”

In sum, we appreciate your encouraging comments and your efforts of providing us with a set of detailed and helpful suggestions in this review. It is clear you have spent a significant amount of time and effort in offering such high-quality feedback. We believe that your suggestions have helped push our thinking further in terms of clarity and depth of contribution. Thank you so much!

Reviewer 5 Report

Comments and Suggestions for Authors

Figure 1 was moved after the literature review, the study objectives were written clearly. In the research method, sampling techniques and data analysis techniques have not been written. In hypothesis testing, there is no coefficient value for the influence between variables, either directly or indirectly

Author Response

Comments 1: Figure 1 was moved after the literature review, the study objectives were written clearly. In the research method, sampling techniques and data analysis techniques have not been written. In hypothesis testing, there is no coefficient value for the influence between variables, either directly or indirectly.  

Response 1: Thank you for providing constructive comments to our paper. We are glad that we have been granted an opportunity to revise and resubmit our paper. We have made further revisions to the paper in response to your comments. Below we detail the revisions we have made in response to the significant issues you’ve identified.   

First, Figure 1 was moved after the literature review.

Second, we added data analysis techniques.

Following is the relevant excerpt from the updated manuscript:

“Statistical analysis was conducted using Mplus 7.0 and SPSS 26.0.”

“The study utilized Confirmatory Factor Analysis (CFA) via Mplus 7.0 software to evaluate how distinct the four key variables were empirically. “

“Employing SPSS PROCESS and opting for the fourth model, the analysis unveiled note-worthy findings. It demonstrated that moral reflectiveness played a significant mediating role in the relationship between mindful leadership and employee green creativity”

“By utilizing the SPSS PROCESS tool and selecting the fourth model, the analysis revealed significant mediation effects of environmental passion on the relationship between mind-ful leadership and employee green creativity”

“The serial mediation model, linking mindful leadership with employee green creativity, was tested using Model 6 of SPSS PROCESS, a method advocated by Hayes (2022) [83] for its comprehensive evaluation of mediating pathways in statistical analyses.”

Third, in hypothesis testing Section, as shown in Table 4 and 5, the coefficient value between mindful leadership and moral reflectiveness was β = .30, p < .001. Moral reflectiveness mediated the association between mindful leadership and Employee green creativity (indirect effect = .04, SE = .02, 95% CI= [.01, .10]), as did environmental passion (indirect effect = .07, SE = .04, 95% CI = [.01, .15]). The results also supported the serial mediating effect (indirect effect = .01, SE = .01, 95% CI [.001, .03]).

Reviewer 6 Report

Comments and Suggestions for Authors

Thank you very much for the opportunity to read and revise this manuscript addressing the mediating effect of moral reflectiveness and environmental passion in the association between mindful leadership and employee green creativity.

Overall, I appreciate the efforts provided by authors to carry out this research on an interesting and present-day topic, such as green creativity. Nonetheless, I have some concerns about this manuscript. I hope that my suggestions can improve the overall quality of this work. Based on my own reading, I believe that the pre-intro (pp. 1-3) should be streamlined, allowing the reader to get immediately the main aims of the study. Some information included in this section could be included in the literature review section. In addition, I believe that the Literature review section is poorly drafted. I suggest improving this section with a deep definition of the research variables, also describing the main theoretical and descriptive frameworks. For instance, I suggest improving the evidence about the relationship between creativity facets and eco-green behaviours focusing on both individual and organizational perspectives. To this end, I suggest including and referring to the following works:

Shah, S. H. A., Fahlevi, M., Rahman, E. Z., Akram, M., Jamshed, K., Aljuaid, M., & Abbas, J. (2023). Impact of green servant leadership in Pakistani small and medium enterprises: bridging pro-environmental behaviour through environmental passion and climate for green creativity. Sustainability, 15(20), 14747.

Giancola, M., Palmiero, M., & D'Amico, S. (2023). The green adolescent: The joint contribution of personality and divergent thinking in shaping pro-environmental behaviours. Journal of Cleaner Production, 417, 138083.

Wesselink, R., Blok, V., & Ringersma, J. (2017). Pro-environmental behaviour in the workplace and the role of managers and organisation. Journal of Cleaner Production, 168, 1679-1687.

As for the method section, I have some concerns about the sample. First, the author should better disclose the sampling strategy. Then, I wonder if the authors planned a sample estimation. Did the authors perform an a priori power analysis in order to establish the minimum sample size? If so, please provide the parameters. In addition, I suggest performing a post-hoc power analysis in order to verify the power reach by the mediation model. If the mediation is underpowered, please collect more data.  

In addition, I suggest improving the quality of the Figure 1. In particular, please ensure that the parenthesis is in the same line as the numbers. In addition, if the authors would like to provide a general picture of the model that they would advance with all research variables, I suggest excluding the statistics. Then, a figure with all path coefficients, direct, indirect, and total effects should be included in the results section. This allows the reader to read the results easily.

Thank you.

Comments on the Quality of English Language

Based on my reading minor editing of English is necessary.

Author Response

Comments 1: Thank you very much for the opportunity to read and revise this manuscript addressing the mediating effect of moral reflectiveness and environmental passion in the association between mindful leadership and employee green creativity.

Overall, I appreciate the efforts provided by authors to carry out this research on an interesting and present-day topic, such as green creativity. Nonetheless, I have some concerns about this manuscript. I hope that my suggestions can improve the overall quality of this work. Based on my own reading, I believe that the pre-intro (pp. 1-3) should be streamlined, allowing the reader to get immediately the main aims of the study. Some information included in this section could be included in the literature review section.

Response 1: Thank you for providing constructive comments to our paper. We are glad that we have been granted an opportunity to revise and resubmit our paper. We have made further revisions to the paper in response to your comments. Below we detail the revisions we have made in response to the significant issues you’ve identified.   We add a general description of the research question in the introduction. Then we rearranged the logic of the department. Finally, we hired a language Polish agency to fix the language. Attached is a proof of language Polish. Following is the relevant excerpt from the updated manuscript (revised sentences are marked by red color, line 81-87):

“Manufacturing firms compete on the global market, and depending on the sector of manufacturing, greening of the production process is highly dependent on technological innovations and international agreements. This paper focuses on the role of the leadership style of Chinese manufacturing firms on employee attitudes (or behaviors) in the context of green creativity. In addition to technological developments, this aspect has the potential for additional value in the greening of production within companies. We aim to explore why and how mindful leadership influences employee green creativity.”

Comments 2: In addition, I believe that the Literature review section is poorly drafted. I suggest improving this section with a deep definition of the research variables, also describing the main theoretical and descriptive frameworks. For instance, I suggest improving the evidence about the relationship between creativity facets and eco-green behaviours focusing on both individual and organizational perspectives. To this end, I suggest including and referring to the following works:

Shah, S. H. A., Fahlevi, M., Rahman, E. Z., Akram, M., Jamshed, K., Aljuaid, M., & Abbas, J. (2023). Impact of green servant leadership in Pakistani small and medium enterprises: bridging pro-environmental behaviour through environmental passion and climate for green creativity. Sustainability, 15(20), 14747.

Giancola, M., Palmiero, M., & D'Amico, S. (2023). The green adolescent: The joint contribution of personality and divergent thinking in shaping pro-environmental behaviours. Journal of Cleaner Production, 417, 138083.

Wesselink, R., Blok, V., & Ringersma, J. (2017). Pro-environmental behaviour in the workplace and the role of managers and organisation. Journal of Cleaner Production, 168, 1679-1687.

Response 2: Thanks for your comment and recommendation. First, definition of the research variables all in introduction. Second, we add more content and we rewrote introduction and Literature review section part and use these references. Following is the relevant excerpt from the updated manuscript (revised sentences are marked by red color, line 174-191):

“First, mindful leadership involves consciously focusing on the present moment and responding intentionally to situations [48]. This approach can greatly benefit employees by encouraging them to cultivate mindfulness themselves. According to SIP theory [36], leaders are perceived as important sources of information because of their significant social status and influence within the organization. Employees will perceive leaders’ ideas, demeanor, and conduct according to this information, which forms the foundation of the employees’ understanding and actions [49]. Mindful leaders’ openness to change signals to employees the importance of adapting to environmental shifts, which fosters a shared understanding of environmental preservation [34, 47]. As a result, employees feel inspired to participate in actions that support corporate environmental initiatives in the workplace [50]. Compared with environmental conservation behaviors, employee green creativity emphasizes the generation of novel ideas, solutions, and approaches that are environ-mentally friendly [51,52,53]. It focuses on fostering innovative thinking and creativity to develop strategies, products, and processes that contribute to environmental sustainability and address environmental challenges effectively [51]. Research has found that mindful-ness can improve individual attention to new stimuli, broaden perspectives, and foster a deeper understanding of complexity within dynamic environments, all of which improves employee creative outcomes [54,55].”

Comments 3:  As for the method section, I have some concerns about the sample. First, the author should better disclose the sampling strategy. Then, I wonder if the authors planned a sample estimation. Did the authors perform an a priori power analysis in order to establish the minimum sample size? If so, please provide the parameters. In addition, I suggest performing a post-hoc power analysis in order to verify the power reach by the mediation model. If the mediation is underpowered, please collect more data. 

Response 3: Thanks for pointing this out and providing us an opportunity to clarify your concern. We are sorry for not mentioning the sample estimation in our data collection. Actually, we estimate the sample size according to Cochran (1977)’s formula: .  refers to the minimum sample size. s refers to estimation of variance. t refers to the t value at a level. d refers to acceptable margin of error.  = Therefore, the lowest sample size for our study is 118. We rewrote this part to show a more complete overview of sample and also rewrite the limitation to note this. Following is the relevant excerpt from the updated manuscript (also see in line 338-360 and line 621-623):

“This research utilized a questionnaire-based survey approach. Questionnaire distribution and retrieval spanned from June 2024 to July 2024. Two waves of data were gathered from four manufacturing firms in China and covered machinery manufacturing, the automobile industry, and the fertilizer sector. We collaborated with human resource man-agers to compile a list of 300 randomly selected employees from a pool of over 600 employees and their respective supervisors. All participants were full-time working adults who participated online. Web-based questionnaires provide rapid distribution and extensive geographic coverage, mitigating the geographic constraints typically associated with traditional survey methods [78].

 Various tactics were employed to address the issue of common method bias, as out-lined by Podsakoff et al. (2012) [79]. First, we utilized established and validated scales from existing literature. Second, prior to commencing the formal inquiry, we provided thorough guidance on questionnaire completion, stressing the importance of confidentiality and highlighting the academic nature of the survey. Third, we adopted a multisource approach. Employees reported individual-level variables such as mindful leadership, moral reflectiveness, and environmental passion, while data relating to employee green creativity were provided by employees’ immediate leaders.

A time-lagged survey-based design was used to examine our research model. In wave 1, we assessed several control variables—namely, age, gender, education, and job tenure—alongside mindful leadership, moral reflectiveness, and environmental passion. These variables were self-evaluated by the participating employees.

A total of 300 questionnaires were disseminated among participants, and upon completion of the survey period, 290 completed questionnaires were collected.’’

“Future research should employ a broader-scale survey, encompassing a more diverse array of manufacturing firms to authenticate and enhance our preliminary observations.”

Comments 4:   In addition, I suggest improving the quality of the Figure 1. In particular, please ensure that the parenthesis is in the same line as the numbers. In addition, if the authors would like to provide a general picture of the model that they would advance with all research variables, I suggest excluding the statistics. Then, a figure with all path coefficients, direct, indirect, and total effects should be included in the results section. This allows the reader to read the results easily.

Response 4: Thanks for this comment. First, we put figure 1 after Literature Review. Then, we improve the quality of the Figure 1.

.25[.13, .37]

Figure 1: Theoretical model

In sum, we appreciate your encouraging comments and your efforts of providing us with a set of detailed and helpful suggestions in this review. It is clear you have spent a significant amount of time and effort in offering such high-quality feedback. We believe that your suggestions have helped push our thinking further in terms of clarity and depth of contribution. Thank you so much!

Round 2

Reviewer 6 Report

Comments and Suggestions for Authors

The authors addressed all points mentioned in the previous round of review. However, as the authors only used a figure with statistics to summarize their results, I recommend including Figure 1 in the results section. After that, the manuscript is suitable for publication.

Comments on the Quality of English Language

Minor editing of English language required.

Author Response

Comments 1: The authors addressed all points mentioned in the previous round of review. However, as the authors only used a figure with statistics to summarize their results, I recommend including Figure 1 in the results section. After that, the manuscript is suitable for publication.

Response 1: Thank you for providing constructive comments to our paper. In light of your comments, we have refined the contents of Figure 1 as follows(please also see this in PDF and word versions):

Figure 1: Theoretical model

Note:

Direct effect      

Mediation effect 1     

Mediation effect 2

Mediation effect 3

We appreciate your encouraging comments and your efforts of providing us with a set of detailed and helpful suggestions in this review. Thank you so much again!
